# Associations of the objective built environment along the route to school with children's modes of commuting: A multilevel modelling analysis (the SLIC study)

**Lander S. M. M. Bosch**[1]*, **Jonathan C. K. Wells**[2], **Sooky Lum**[3], **Alice M. Reid**[1]

**1** Department of Geography, University of Cambridge, Cambridge, England, United Kingdom, **2** UCL Great Ormond Street Institute of Child Health, London, England, United Kingdom, **3** Respiratory, Critical Care & Anaesthesia Section, London, England, United Kingdom

* lsmmb2@cam.ac.uk

## Abstract

As active commuting levels continue to decline among primary schoolchildren, evidence about which built environmental characteristics influence walking or cycling to school remains inconclusive and is strongly context-dependent. This study aimed to identify the objective built environmental drivers of, and barriers to, active commuting to school for a multi-ethnic sample of 1,889 healthy primary schoolchildren (aged 5–11) in London, UK. Using cross-sectional multilevel ordered logistic regression modelling, supported by the spatial exploration of built environmental characteristics through cartography, the objective built environment was shown to be strongly implicated in children's commuting behaviour. In line with earlier research, proximity to school emerged as the prime variable associated with the choice for active commuting. However, other elements of the urban form were also significantly associated with children's use of active or passive modes of transport. High levels of accidents, crime and air pollution along the route to school were independently correlated with a lower likelihood of children walking or cycling to school. Higher average and minimum walkability and higher average densities of convenience stores along the way were independently linked to higher odds of active commuting. The significance of the relations for crime, air pollution and walkability disappeared in the fully-adjusted model including all built environmental variables. In contrast, relationships with proximity, traffic danger and the food environment were maintained in this comprehensive model. Black children, pupils with obesity, younger participants and those from high socioeconomic families were less likely to actively commute to school. There is thus a particular need to ensure that roads with high volumes of actively commuting children are kept safe and clean, and children's exposure to unhealthy food options along the way is limited. Moreover, as short commuting distances are strongly correlated with walking or cycling, providing high-quality education near residential areas might incite active transport to school.

**Data Availability Statement:** All relevant data are within the manuscript and its Supporting Information files.

**Funding:** SL and the SLIC study were supported by the Wellcome Trust [WT094129MA]. URL: https://wellcome.ac.uk/ LB receives funding for his doctoral study from Gonville and Caius College, University of Cambridge, UK (no grant number). URL: https://www.cai.cam.ac.uk/ LB is also supported by the UK Economic and Social Research Council [ES/J500033/1]. URL: https://www.esrcdtp.group.cam.ac.uk/ The funders had no role in study design, data collection and analysis, decision to publish, or preparation of the manuscript.

**Competing interests:** The authors have declared that no competing interests exist.

**Abbreviations:** BAME, Black, Asian and Minority Ethnic; FEAT, Food Environment Assessment Tool; FMI, Fat Mass Index; LSOA, Lower Super Output Area; SLIC, Size and Lung function In Children; WHO, World Health Organization.

# Introduction

## 1.1 Active commuting, physical activity and health

Children's physical activity levels are dwindling in the UK and around the world [1]. Simultaneously, rates of childhood obesity continue to rise globally [2]. This double issue is particularly problematic in urban environments. In London, for example, only a quarter of children meet current physical activity guidelines [3]. Moreover, by the end of primary school, nearly forty percent of children are overweight or obese [4]. Increased physical activity is linked to a wide range of physical and mental health benefits [5,6]. To maintain and improve child health, the World Health Organization (WHO) recommends at least 60 daily minutes of moderate-to-vigorous physical activity for children aged 5–17 [7]. Active school commuting holds great potential as a source of daily physical activity for children, and has been associated with higher activity levels throughout the day [8,9]. Whilst evidence is still scarce, active commuting has also been linked to lower fat mass in children [10]. However, active mobility interventions tend to have small effect sizes [11]. It is therefore important to identify drivers of, and barriers to, children's propensity to walk, scoot or cycle to school, to devise policy interventions that can raise levels of active mobility.

## 1.2 Active commuting and the built environment

Due to their higher physical activity levels and rapid development, children are strongly influenced by the variety of built environments to which they are exposed [12]. The built environment is also heavily implicated in transport-related decisions [13]. Six objectively measurable built environmental variables are frequently suggested to influence children's commuting mode choices, or require further clarification: school proximity; traffic risk; personal risk; air pollution; walkability; and the food environment. The first of these variables, proximity, has been most consistently associated with children's mode of commuting [13].

Secondly, parental and child concerns about traffic safety may impact commuting decisions, as may personal safety risks, the third variable. Risk perceptions, particularly those of parents, are increasingly linked to children's physical activity and active travel [9, 14]. However, consistent evidence is lacking on whether objective accident and crime rates significantly influence commuting decisions [9].

Next, while the adverse consequences of air pollution on population health have been frequently reported [15], its effect on child physical activity is largely unknown, especially in a European context [16]. Exploring this association is crucial, as UK research highlights that children walking to school are exposed to higher pollution levels than passive commuters [17].

To capture the joint impact of built environmental characteristics on stimulating active transport, the concept of 'walkability' has been introduced [18]. Conventionally, information on land use mix, street connectivity and residential density is combined in a composite index [19]. Walkability has been shown to be a consistent correlate of adult walking and cycling [20]. However, evidence for children is much sparser [12, 21].

Finally, the food environment is widely recognized to be an important driver of weight gain [22]. Commuters are directly exposed to the foodscapes in their surroundings [23], and this has been preliminarily linked to children's activity levels [24]. The potential link between the food environment and children's transport to school should therefore be explored.

## 1.3 Study aim

The present research aimed to study the associations of the complex web of built environmental characteristics with children's mode of commuting to school for a multi-ethnic urban

sample of UK primary schoolchildren. It achieved this aim through the combination of multi-level modelling analyses and cartography.

## Materials and methods

### 2.1 Sample

Cross-sectional analyses were performed on data collected between December 2010 and June 2013 for 1,889 primary schoolchildren in good health aged 5–11, attending thirteen schools in London. These children participated in the Size and Lung function In Children (SLIC) study, carried out at the Great Ormond Street Institute of Child Health, University College London, UK [25]. 53.7% of this sample of SLIC children were female. About a quarter were aged five to six (24.6%), 38.1% were seven to eight years old, and the remaining 37.3% were aged nine to eleven. The purposeful sampling of schools based on their diverging geographical location, education performance and ethnic mix guaranteed the inclusion of a diverse, representative sample of London primary schoolchildren. Further details are provided elsewhere [25, 26]. Ethical approval for the SLIC study was obtained from the London-Hampstead research ethics committee (REC: 10/H0720/53). Parental written consent and child verbal assent were obtained prior to assessments for all participants to the SLIC study. Approval for secondary data analyses was obtained from the research ethics committee at the Department of Geography, University of Cambridge, UK (Ethics Assessment Number 698).

### 2.2 Variables

**2.2.1 Response variable–mode of commuting.**   Using a questionnaire, SLIC children and their parents/guardians were independently asked about the dominant mode of school transport [25,26]. Their response was classified as 'active commuting' if the child predominantly walked or cycled. Where car, bus or underground were dominant, this was labelled 'passive commuting'. If both active and passive modes of transport contributed significantly, this was classified as 'mixed commuting'. Where parent and child responses differed, this disagreement was interpreted as an indication of variable or mixed commuting. Hence, this group was added to the 'mixed' category.

**2.2.2 Predictors–built environmental data.**   Values for the six built environmental variable values described above were calculated along the shortest route between the place of residence of each SLIC child and the school she or he attended, using ArcMap 10.5.1 (ESRI 2017, Redlands, CA, USA). The location of the SLIC child's home, obtained through the parental questionnaire, was included as the centroid of their 2011 Output Area of residence. Output Areas are the smallest spatial units meeting the confidentiality threshold (containing minimum 40 households) for which UK census data are available [27]. With 95% of Output Areas containing between 79 and 189 households [27], their centroids can be assumed to reasonably approximate the actual location of SLIC children's homes. Distance to school was calculated along the shortest route (in metres) between this centroid and the school the child attended. Commuting distances were subdivided into four categories: <500.0 metres, 500.0–999.9 metres, 1,000.0–1,499.9 metres and ≥1,500.0 metres. 1,500 metres is often considered to be the maximum walkable distance to school [28].

For the other built environmental variables, two values were calculated. On the one hand, the average value across all administrative units traversed by children along the shortest way to school was computed. On the other, the most extreme, 'worst-case' value encountered by children during their commute was included, often assumed to be more impactful in transport decisions [29].

Traffic risk was measured by the 2011 rates of 'accidents with injury' in the Lower Super Output Areas (LSOAs) SLIC children crossed along their way to school. LSOAs were selected as they balance the detail provided by smaller spatial units with the statistical reliability of larger, more populated areas. Accident data were collected by the UK Department for Transport [30] and categorized as <20.0, 20.0–39.9 or ≥40.0 accidents with injury per 10,000 inhabitants.

Personal risk was calculated using 2010/2011 crime rates, based on the total number of notifiable offences in the LSOAs children passed. These data were collected by SafeStats London per financial year, and made available in the 2011 London LSOA Atlas [31]. They are subdivided into three dimensionless categories: <90.0, 90.0–109.9 and ≥110.0.

The Combined Emissions Index available in the same database was used as the measure for air pollution [31]. Nitrogen Oxide, Nitrogen Dioxide and Particulate Matter ($PM_{10}$) concentrations were combined to assign an overall air quality score to each LSOA. As pollutant concentrations are closely related to local motorized vehicle exhaust [32], this also served as a proxy of traffic density. This dimensionless index was subdivided into three categories: <90.0, 90.0–109.9 and ≥110.0.

Stockton and colleagues [19] computed a walkability index for London Output Areas, combining residential dwelling density, density of three- or more-way junctions and land use mix. The walkability quintile scores for the 2011 Output Areas were selected as the walkability measure.

The food environment was included as the categorized density of convenience stores in a one-mile radius around the postcodes the child crossed during the commute (≤20, 21–50, 51–80 or >80 stores/mile$^2$). These data were collected as part of the Food Environment Assessment Tool (FEAT) project [33]. The earliest available data, from 2014, were used.

**2.2.3 Potential confounders.** Information on children's age (in years), sex and school attended was collected during school visits. Ethnicity data were collected via the parental questionnaire [25, 26]. Children were assigned to one of three ethnic groups: black (African/Caribbean ancestry), South Asian (ancestry from the Indian subcontinent) or white/other (European/other/mixed ancestry). Body composition might be associated with children's commuting decisions, although the directionality of relations cannot be causally established in cross-sectional models. Sample-specific percentiles of children's age- and sex-adjusted Fat Mass Index (FMI) were calculated using data on fat mass (estimated through bio-electrical impedance analysis using standing instrumentation; Tanita BC418, Tanita Corporation, UK) adjusting for the square of height. As fat mass data were skewed, these were natural log-transformed prior to the calculation of z-scores. Similar to Centers for Disease Control and Prevention guidelines [34], SLIC children were assigned to one of four weight status categories, using the 5th, 85th and 95th percentiles as class boundaries (underweight, normal weight, overweight or obese). Data on family socioeconomic status were also gathered via the questionnaire [25]. The three included variables are: Family Affluence Scale category (low, intermediate or high), whether or not the child received free school lunches, and car ownership (zero to two cars). Finally, neighbourhood deprivation was captured by the 2010 Index of Multiple Deprivation score for the postcode of residence (categorized as low, intermediate or high deprivation). Detailed information on these measures has been published previously [10].

## 2.3 Analyses

Data visualisation sheds light on the spatial distribution of potential drivers to, and barriers of, children's active commuting across London. Hence, firstly, the six built environmental characteristics and two potential confounders for which London-wide data were available, ethnicity

and deprivation, were mapped in ArcMap 10.5.1. For convenience store density, accurate data were only provided for the postcode areas where SLIC children resided. For other areas, Borough-level data are shown. 32 Boroughs, or local authorities, make up Greater London, with an average population of around 255,000 inhabitants at the time of the 2011 population census [35]. As no London-wide FMI data were available, this variable could not be included in the cartographic analyses.

Statistical analyses were carried out in Stata 15 (StataCorp 2017, College Station, TX, USA). Performing multilevel modelling analysis on two levels avoids violating the assumption of independence of observations due to the nested data structure [36, 37]. The first level captures individual and family characteristics through the inclusion of children's age, sex, ethnicity and family and neighbourhood socioeconomic status. The second level then corrects for the higher likelihood of similar home and school environments of children attending the same school by grouping them by school. The response variable, mode of commuting, can be interpreted as an ordinal categorical variable, whose outcomes can be ranked from passive commuting, involving the least physical activity, through mixed commuting, to active commuting, entailing the highest activity levels. This resulted in the design of a mixed-effects two-level ordered logistic regression model [38].

Two sets of models were designed. Firstly, associations between the average and extreme value for each individual built environmental characteristic and SLIC children's likelihood of active commuting to school are presented, fully corrected for potential confounders. Secondly, the results of a comprehensive, fully-adjusted model including all six built environmental measures are shown and discussed. The choice to include either the average or extreme value for each built environmental variable was made depending on the strength and significance of their independent associations with commuting in the individual models.

In practice, these models were estimated using Stata's mixed-effects ordered logistic regression ('meologit') command function. Children's mode of commuting was the ordinal response variable in all models. The built environmental variables were included as first-level explanatory variables, individually in the first set of models and combined in the second set, together with the full range of potential confounders. Following their categorization, all explanatory built environmental variables were treated as factor variables in the models (using the 'i.' command in Stata), as were all potential confounders with exception of age, the latter being considered a continuous variable. The lowest category for each built environmental characteristic was consistently set as reference. For potential confounders, the selected reference categories were female sex, white/other ethnicity, normal fat mass, low family affluence, no receipt of free school lunches, no family car ownership and low Index of Multiple Deprivation. On the second level of the multilevel analyses, the identification number of the schools was then included in all models as the group variable.

Results are shown as Odds Ratios with 95% Confidence Intervals for active commuting in comparison to mixed or passive commuting. Statistical significance was set at p<0.05.

## Results

### Cartography and descriptive statistics

The sample data are made available in *S1 Table*. SLIC children resided primarily in the Borough where their school was located, mainly in the northern part of London (*Fig 1*). Three schools were in the Boroughs of Newham and Enfield, two in Southwark and Brent, and one each in Haringey, Hackney and Harrow. 46.7% of children predominantly commuted actively to school, and 31.7% used mainly passive means of transportation (*Table 1*). 21.6% belonged

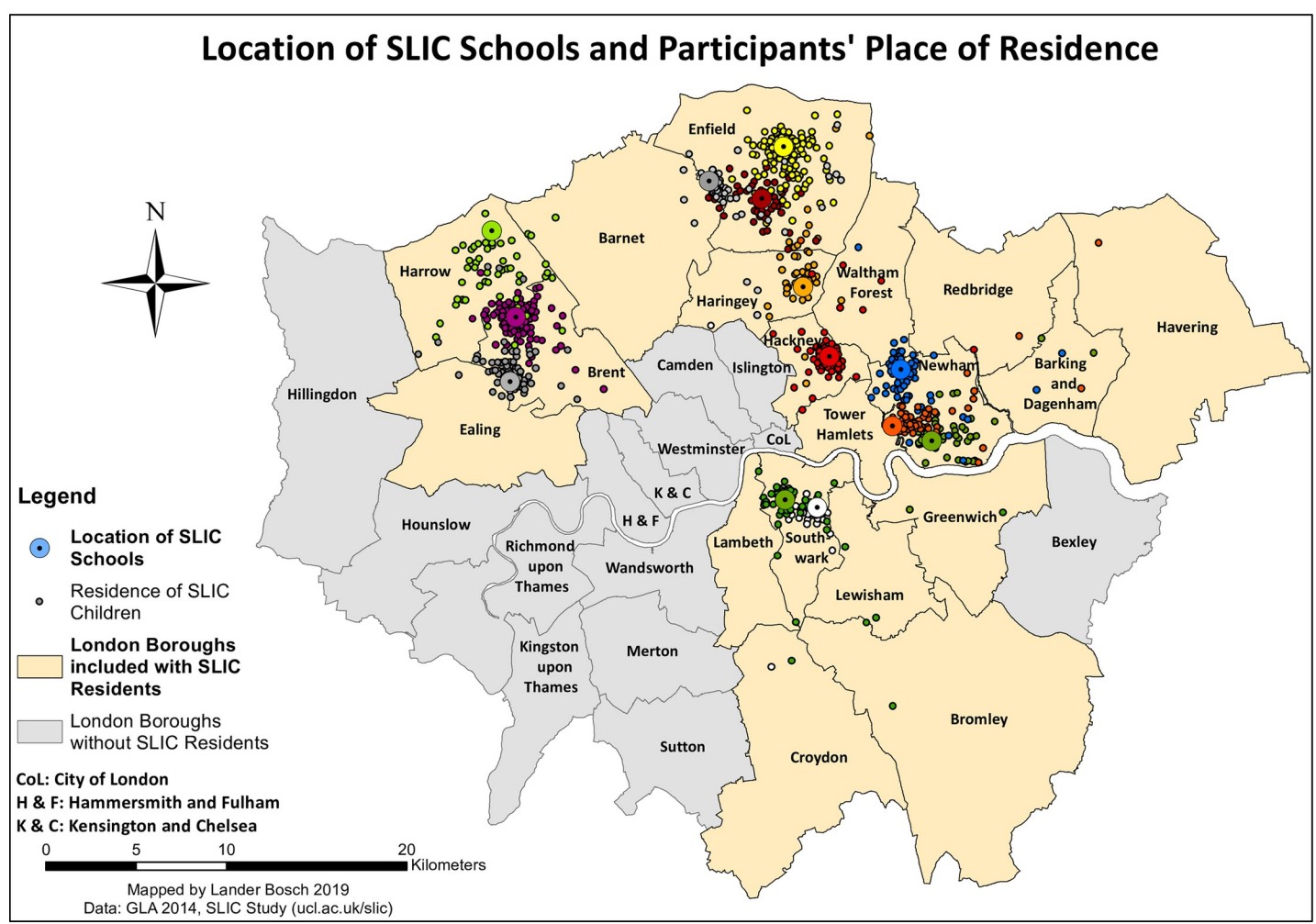

**Fig 1. Proximity to school, data © see acknowledgements.**

to the mixed/disagreement category. One of ten children resided within 500 metres of school, while for 41.7% of children, the shortest distance to school was over 1,500 metres.

Traffic risk varied strongly across London, though consistently high accident rates were found in the city centre (*Fig 2*). Children, especially those with longer commuting distances, were thus likely to be exposed to strongly diverging degrees of risk along the way. The descriptive statistics corroborate this observation: the average traffic risk along the route fell in the lowest category for 31.3% of children, compared to 18.9% in the highest category. However, 62.5% of participants were confronted with at least one LSOA segment with the highest risk.

Personal risk in London showed a clear spatial pattern (*Fig 3*). Crime rates were high along quasi-perpendicular axes running from west to east and north to south, converging in Central London. SLIC children residing and attending school towards the edges of the city (in Enfield and Harrow) were confronted with lower personal risk than those closer to the centre. 48.7% were exposed to low average levels of crime along the commuting route. However, over two-thirds of pupils (69.1%) passed through at least one LSOA in the highest crime rate category.

Air pollution showed a distinct radial pattern (*Fig 4*), with the highest concentrations of airborne toxins found in the central Boroughs. Local pollution hotspots elsewhere, often following linear patterns, can be linked to major traffic arteries or, in West London, the presence of

**Table 1. Commuting and built environmental measures for the sample of SLIC children in the current study.**

| Variable | Number of SLIC children | Percentage (%) |
|---|---|---|
| **Dominant mode of commuting** | | |
| *Active* | 883 | 46.7 |
| *Passive* | 598 | 31.7 |
| *Mixed & Disagreement* | 408 | 21.6 |
| **Proximity to School–Network (m)** | | |
| *< 500.0* | 190 | 10.1 |
| *500.0–999.9* | 457 | 24.2 |
| *1,000.0–1,499.9* | 454 | 24.0 |
| *≥ 1,500.0* | 788 | 41.7 |
| ***Average Traffic Risk along Route*** | | |
| *Average accident rate per $10^4$ inhabitants crossed* | | |
| *<20.0* | 591 | 31.3 |
| *20.0–39.9* | 940 | 49.8 |
| *≥ 40.0* | 358 | 18.9 |
| ***Maximum Traffic Risk along Route*** | | |
| *Highest accident rate per $10^4$ inhabitants crossed* | | |
| *<20.0* | 151 | 8.0 |
| *20.0–39.9* | 557 | 29.5 |
| *≥ 40.0* | 1 181 | 62.5 |
| ***Average Personal Risk along Route*** | | |
| *Average crime rate crossed* | | |
| *<90.0* | 919 | 48.7 |
| *90.0–109.9* | 193 | 10.2 |
| *≥ 110.0* | 777 | 41.1 |
| ***Maximum Personal Risk along Route*** | | |
| *Highest crime rate crossed* | | |
| *<90.0* | 423 | 22.4 |
| *90.0–109.9* | 161 | 8.5 |
| *≥ 110.0* | 1 305 | 69.1 |
| ***Average Air Pollution along Route*** | | |
| *Average Combined Emission Index crossed* | | |
| *<90.0* | 626 | 33.1 |
| *90.0–109.9* | 975 | 51.6 |
| *≥ 110.0* | 288 | 15.3 |
| ***Maximum Air Pollution along Route*** | | |
| *Highest Combined Emission Index crossed* | | |
| *<90.0* | 212 | 11.2 |
| *90.0–109.9* | 983 | 52.0 |
| *≥ 110.0* | 694 | 36.8 |
| ***Average Walkability along Route*** | | |
| *Average quintile crossed* | | |
| *1 (least walkable)* | 182 | 9.6 |
| *2* | 545 | 28.9 |
| *3* | 544 | 28.8 |
| *4* | 526 | 27.8 |
| *5 (Most walkable)* | 92 | 4.9 |
| ***Minimum Walkability along Route*** | | |

*(Continued)*

**Table 1.** (Continued)

| Variable | Number of SLIC children | Percentage (%) |
|---|---|---|
| *Lowest quintile crossed* | | |
| *1 (least walkable)* | 1 226 | 64.9 |
| *2* | 356 | 18.8 |
| *3* | 221 | 11.7 |
| *4* | 79 | 4.2 |
| *5 (Most walkable)* | 7 | 0.4 |
| *Average Food Environment along Route* | | |
| *Average convenience stores/mile$^2$ crossed* | | |
| *$\leq$20* | 78 | 4.1 |
| *21–50* | 1 005 | 53.2 |
| *51–80* | 226 | 12.0 |
| *> 80* | 580 | 30.7 |
| *Maximum Food Environment along Route* | | |
| *Highest convenience stores/mile$^2$ crossed* | | |
| *$\leq$20* | 32 | 1.7 |
| *21–50* | 761 | 40.3 |
| *51–80* | 432 | 22.9 |
| *> 80* | 664 | 35.1 |

Heathrow Airport. 15.3% of children were faced with average pollution levels in the highest category during their commute. However, over a third of pupils (36.8%) were confronted with a maximum Combined Emission Index score of over 110 in at least one LSOA along the route.

Walkability also reduced from the centre of the city to the outskirts (*Fig 5*), with local hotspots in the denser urban cores of Outer London Boroughs. Hence, walkability levels around SLIC participants' homes and schools in Newham, Southwark and Hackney were generally higher than those in Haringey, Brent, Enfield or Harrow. 85.5% of children commuted along roads with an average Walkability Index score in the intermediate second, third or fourth quintiles. However, 64.9% had to cross at least one Output Area in the lowest quintile.

The spatial distribution of convenience stores showed a patchy pattern. Densities were generally highest towards Central London, with hotspots of these outlets across the city (*Fig 6*). SLIC schools and pupils residing in Southwark, Hackney, Newham and Haringey were surrounded by highest convenience stores densities. For only 4.1% of children, the average density on the way to school fell in the lowest category ($\leq$20 stores/mile$^2$), versus nearly a third with over 80 stores/mile$^2$. 35.1% were confronted with these high densities at specific points of their commute.

Further descriptive statistics relating to confounders are provided in *S2 Table*. Just over a quarter of SLIC children were black, and a similar share were South Asian. The black, Asian and minority ethnic (BAME) population was particularly well-represented in west and north-west London, where the SLIC population was predominantly South Asian, and in the south, east and north-east, where SLIC children from minority backgrounds were predominantly black (*Fig 7*). 9.1% and 6.1% of participants were classified as having overweight or obesity based on their within-population FMI score, respectively. About two-thirds of SLIC families had intermediate affluence levels, and a similar share did not receive free school lunches. SLIC children in the northern Boroughs of Harrow and Enfield were more likely to belong to more affluent families (*Fig 8*). A quarter of families did not own a car. Deprivation followed a similar

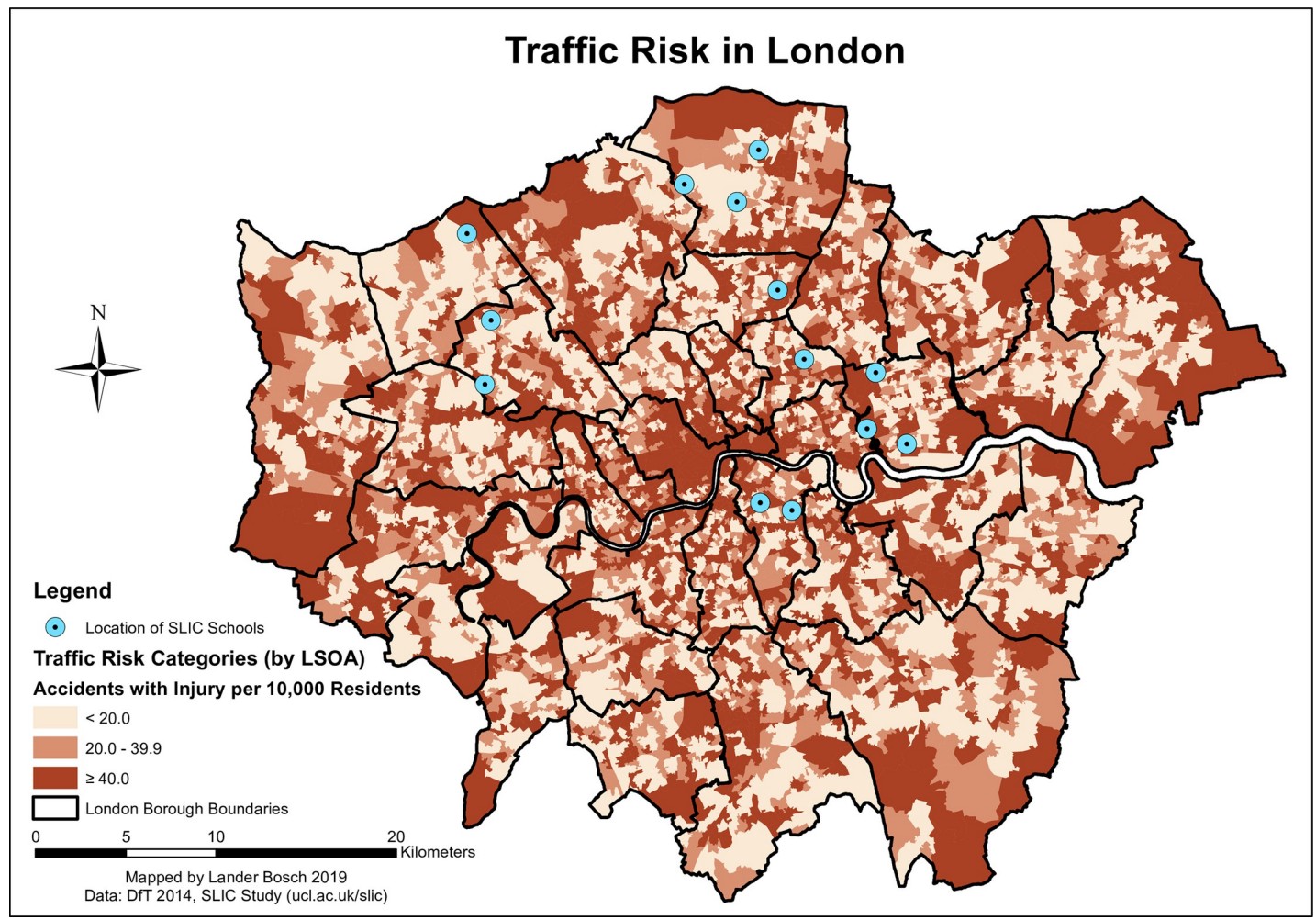

**Fig 2. Traffic risk in London, data © see acknowledgements.**

spatial pattern to crime rates (*Fig 8*). 52.1% of SLIC children resided in LSOAs belonging to the lowest two deprivation quintiles.

## Multilevel models linking built environmental characteristics and commuting to school

The results of the multilevel models assessing the paired associations between each of the six built environmental characteristics individually and SLIC children's mode of commuting to school are shown in *Table 2*. These models were corrected for individual characteristics and family and neighbourhood socioeconomic status, and the associations described remained unchanged in models excluding FMI.

Children living further away from school had significantly lower odds of actively commuting to school compared to those residing within 500 metres. This inverse relationship intensified as commuting distance increased. Both average and extreme rates of traffic accidents along the route to school were significantly associated with SLIC children's commuting mode choices. Participants exposed to higher road risk were significantly less likely to commute actively to school, an effect which became stronger as danger increased. SLIC children who

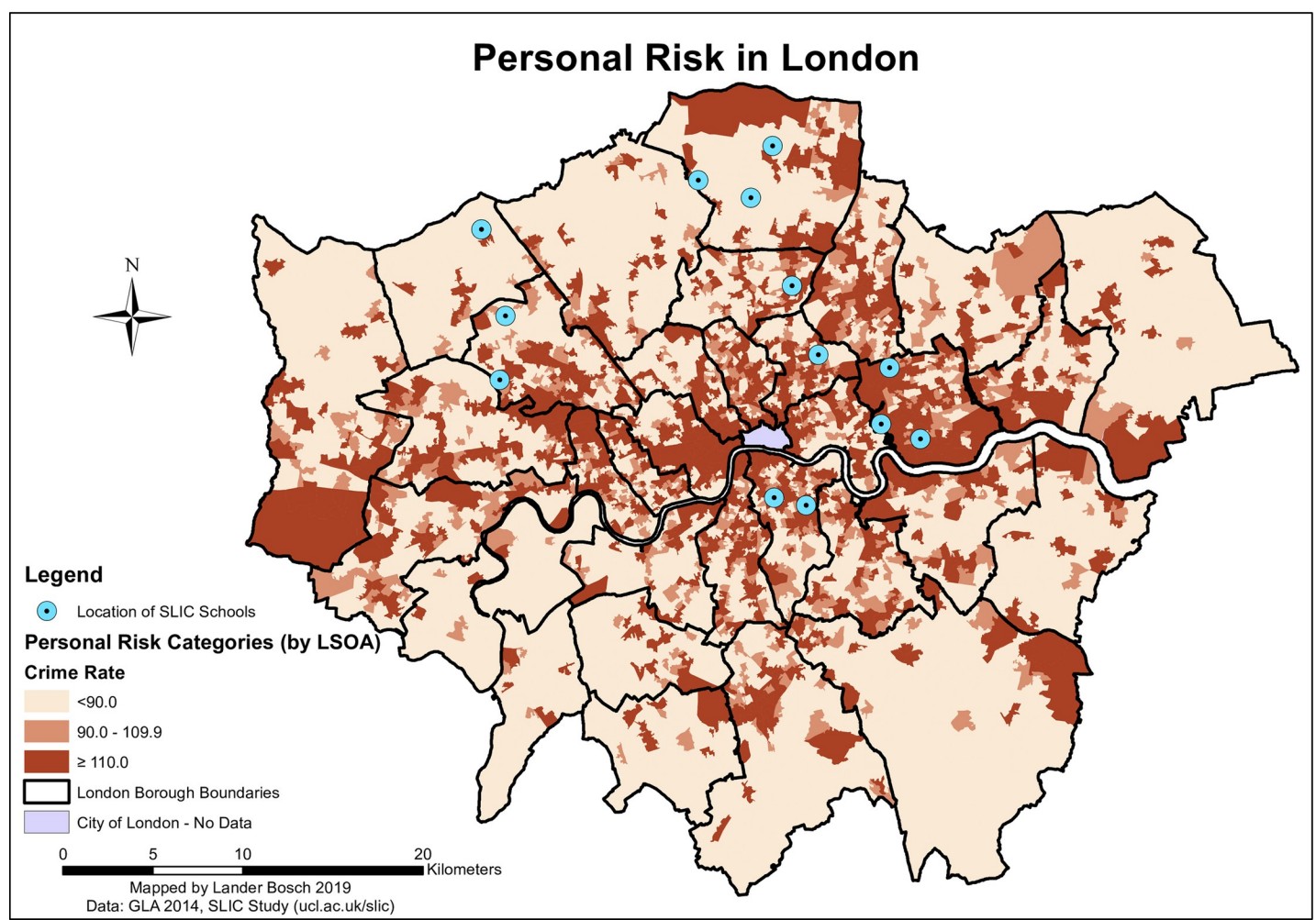

**Fig 3. Personal risk in London, data © see acknowledgements.**

needed to pass through an LSOA with higher extreme crime rates were significantly less likely to actively commute to school. Again, the effect size increased as crime rates rose. No significant associations were found for average personal risk.

In terms of air quality, only maximum pollution levels were significant. Children confronted with higher maxima of airborne pollutant concentrations were less likely to use active modes of transport. Children with the most walkable commutes on average were significantly more likely to walk or cycle to school compared to participants with the least walkable commuting trajectories. Similar observations could be made for walkability extremes, except for the fifth quintile. Finally, an average convenience store density of over 50 outlets/mile$^2$ along the shortest route to school was associated with significantly higher active commuting odds in comparison to a route with the lowest average density. No such associations were found for extreme densities.

Given their significant associations with commuting choices, the extremes of traffic risk, personal risk, air pollution and walkability were retained in the fully corrected model including all built environmental variables and potential confounders. For the food environment, the relation with average convenience store density was stronger. Hence, this average measure was selected. *Table 3* presents the resulting associations.

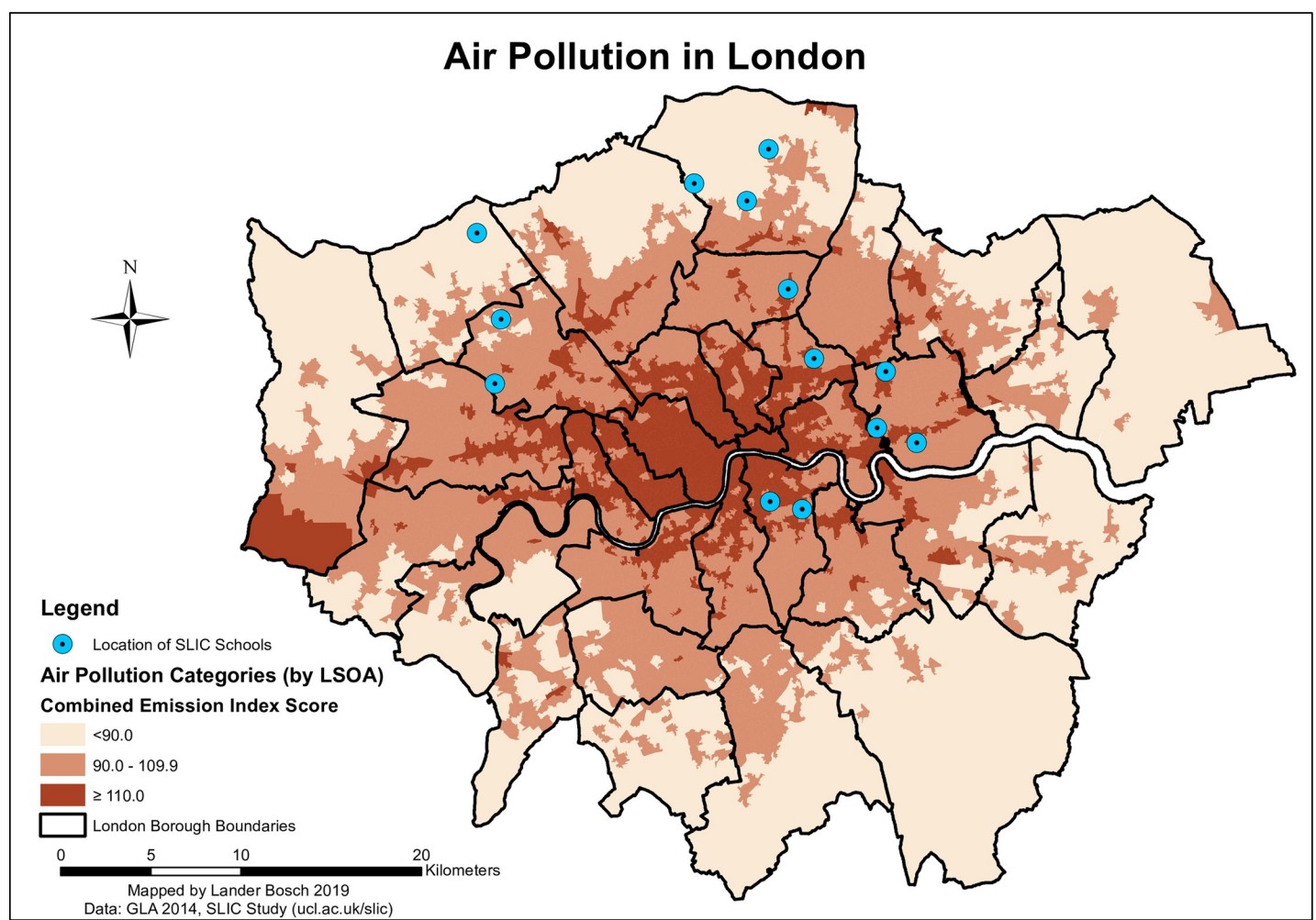

**Fig 4. Air pollution in London, data © see acknowledgements.**

Three variables retained significant associations with commuting mode choices. First, SLIC participants residing further away from school had lower odds of walking or cycling to school. This proximity effect became stronger as distance increased. Secondly, in comparison to children exposed to the lowest accident rates, those confronted with intermediate traffic risk were less likely to actively commute. Finally, children exposed to an average density of 51–80 convenience stores/mile$^2$ had over three times the odds of commuting actively compared to their peers surrounded by the lowest food outlet densities.

Looking at potential confounders, no significant sex differences emerged. Older children were increasingly likely to use active modes of school transport. Compared to the white/other group, black SLIC children had about half the odds of choosing active commuting modes. No such relation emerged for the South Asian group. In comparison to children with normal fat mass, children in the obese category had significantly lower odds of actively commuting. Participants from highly affluent families were significantly less likely to walk or cycle to school in comparison to those from a low-affluence family. Car ownership, and particularly a second car, significantly reduced the odds ratios for active commuting. No significant relation was obtained for neighbourhood deprivation.

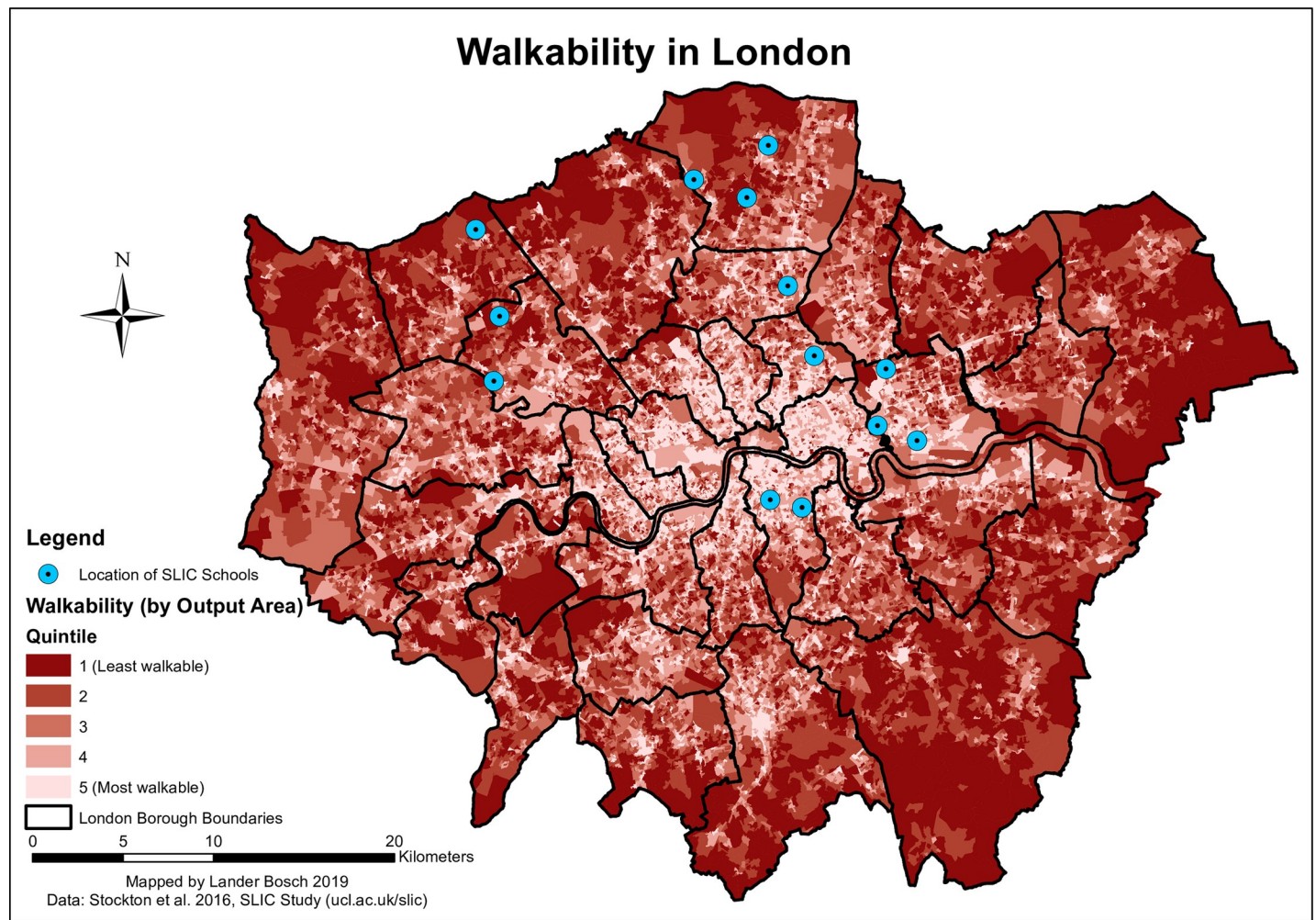

**Fig 5. Walkability in London, data © see acknowledgements.**

## Discussion

This study investigated the associations between objective built environmental characteristics and commuting mode choices for a multi-ethnic sample of UK children participating in the SLIC study. To our knowledge, it is the first to assess the associations of this wide diversity of objective built environmental variables with transport decisions for urban schoolchildren in the UK throughout primary school.

### Associations between built environmental characteristics and modes of commuting to school

Distance to school was consistently and negatively associated with active commuting to school for SLIC children, both in the individual and comprehensive models and irrespective of potential confounders or mediators. The associations were stronger for longer commutes. Prior evidence points to the importance of a limited distance to school in the decision to walk or cycle [13, 39, 40]. The odds ratios for active versus passive or mixed commuting for pupils residing over 1,500 metres from school were just over one-tenth of those living closest to school. Hence the criterion distance for walking to school for children, set around 1.5 kilometres [28, 41],

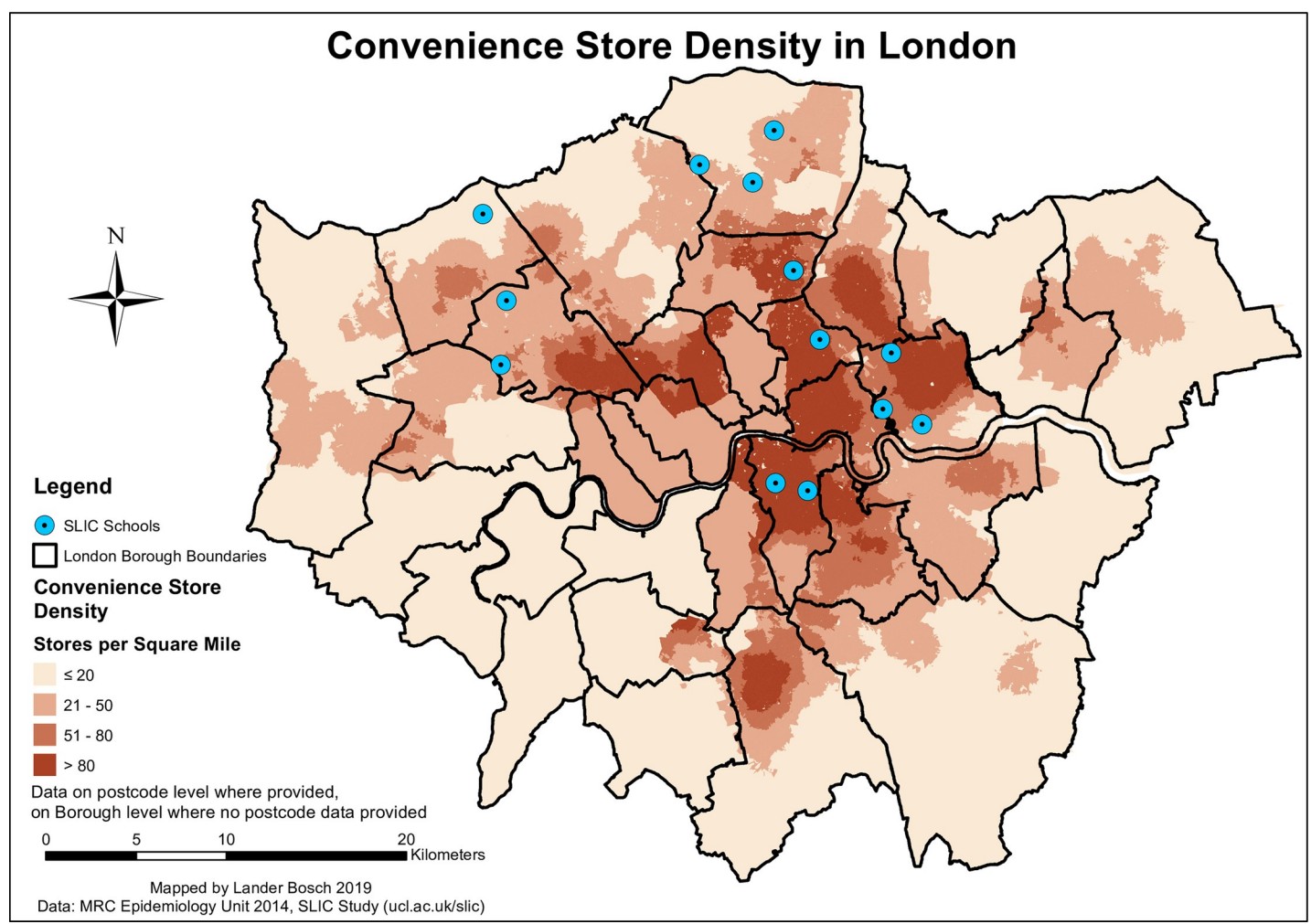

**Fig 6. Convenience store density in London, data © see acknowledgements.**

should be interpreted as a hard barrier to active commuting for this sample of London school-children. Longer distances to school are not only related to longer travel times, but also to increased practical constraints and safety concerns [42]. The cartography demonstrated that longer commutes increase the likelihood that children will encounter disadvantageous environmental conditions along the way. The strategic location of schools within the walkable catchment area of neighbourhoods with a high population share of schoolchildren might thus increase levels of active commuting [41, 42]. Moreover, this might reduce the reliance on free public transport provided to pupils deemed to live too far from the nearest suitable school or commuting along unsafe walking routes. Currently, boundaries for free school transport are set at 3.2 kilometres for children aged 8 or under, and 4.8 kilometres for those aged 8–16 [43].

SLIC children were less likely to actively commute if they crossed unsafe neighbourhoods. In the individual models, both average and extreme accident risk showed significant associations, as did extreme crime rates. The association between unsafe traffic conditions and children's lower odds of active commuting remained significant in the comprehensive model for the group encountering a maximum accident level between 20 and 40 per 10,000 inhabitants. This link between higher exposure to more hazardous road environments and a lower

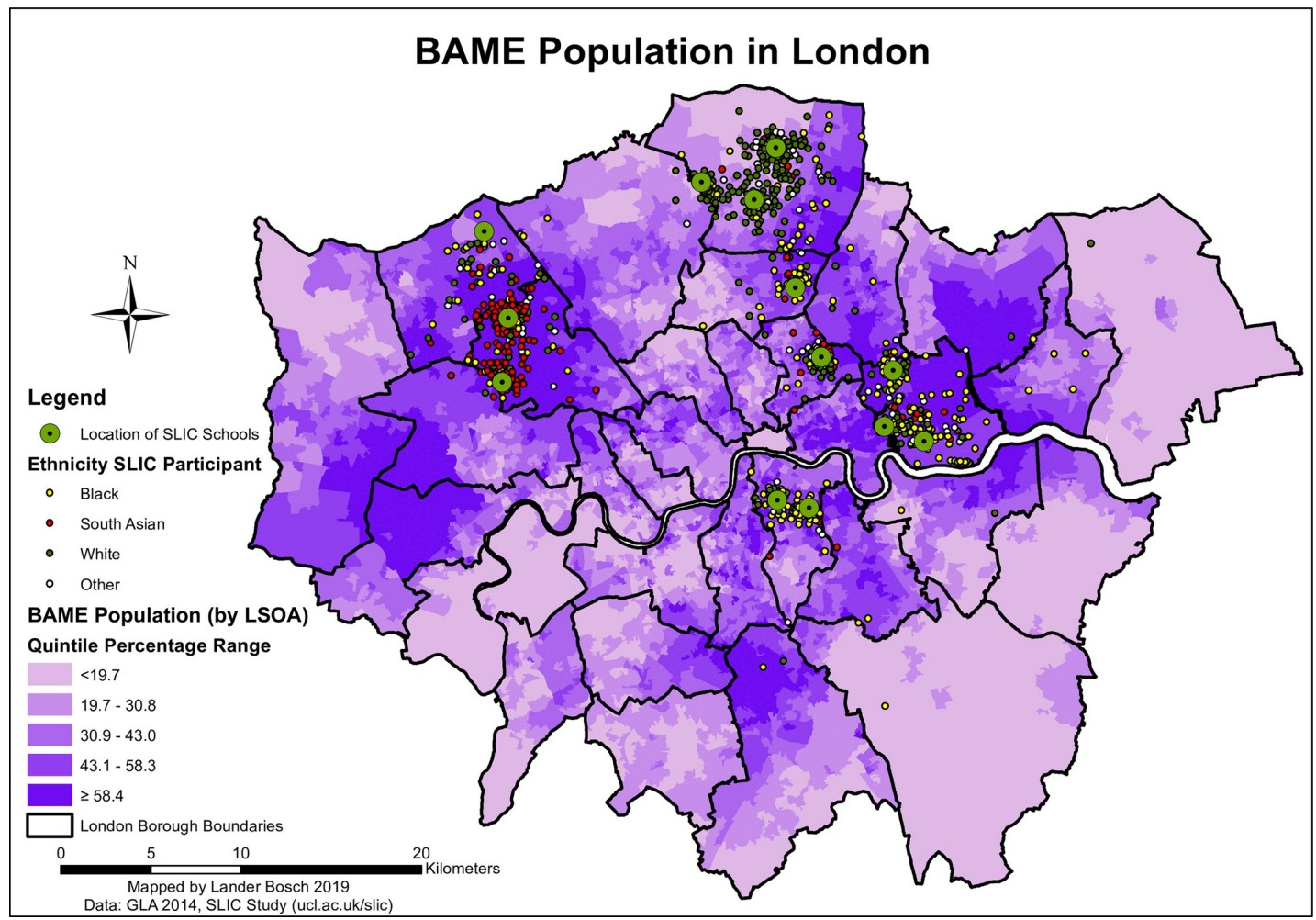

**Fig 7. BAME population in London, data © see acknowledgements.**

likelihood to walk or cycle to school supports earlier research in a Dutch context [44]. It also shows that the 'worst case scenario' along part of the route may act as a particularly strong deterrent to active travel [29]. As actively commuting children are particularly vulnerable road users [17, 45], the need for interventions to reduce the risk of traffic injuries is pressing. The traffic safety map showed that hotspots of risk occur across the city, making this a London-wide concern. Prior research has highlighted that (primarily parental) safety perceptions can be decisive in determining children's activity [46]. However, our findings indicate that objective safety can also be linked to commuting decisions, or, that parental perceptions closely match objective risks. While providing an objectively safe commuting environment is thus pivotal, this might not automatically result in higher levels of active transport if it is not followed by an immediate or longer-term increase in perceived safety [9].

In the individual model, highs of pollutant concentrations were also found to significantly reduce the odds for children to use active transport. Children thus appeared to be deterred by unhealthy levels of air pollution and, as a proxy, of dense traffic en route to school. The maps showed these high concentrations were primarily found in Central London and along major traffic arteries. This finding was not retained in the comprehensive model, perhaps due to the predominance of more immediate road risks over air quality concerns. Nonetheless,

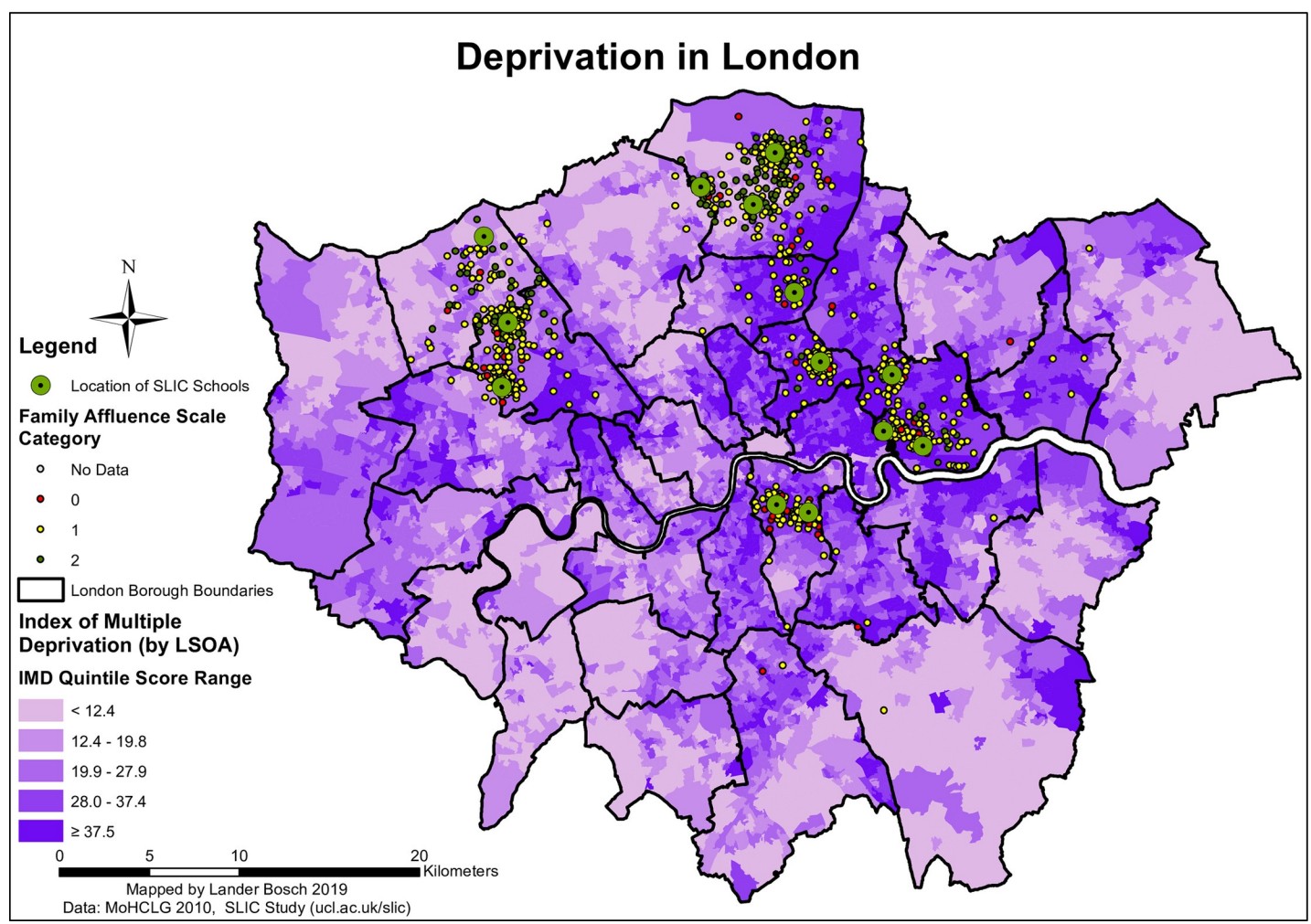

**Fig 8. Deprivation in London, data © see acknowledgements.**

addressing high levels of air pollution is vital, given the disproportional exposure of active commuters to airborne toxins [17].

The results also add to the hitherto equivocal evidence for walkability [12]. Similar to findings for adults, our results for the individual model underline that the combination of high residential density, street connectivity and land use mix, highest in Central London, could stimulate children's physical activity. A similar, positive association was found in three out of five cases elsewhere in Europe [47]. However, the lack of significant associations in the comprehensive model shows that other built environmental characteristics appear to be more influential.

While the food environment is heavily implicated in the childhood obesity epidemic, its potential role in shaping behavioural habits and generating mobility bias has only recently been conceptualized [24, 48]. Here, the individual multilevel model revealed that SLIC children surrounded by 50 convenience stores/mile$^2$ or more on average had higher odds of active commuting than those living in areas with the lowest outlet densities. This significant association was retained in the comprehensive model for participants encountering an average of 51–80 convenience stores/mile$^2$ during the school commute. These children, mainly residing

**Table 2. Associations between individual built environmental measures and SLIC children's active commuting[a].**

| Variable | Odds Ratio [Confidence interval] | p-value |
|---|---|---|
| *Proximity* | | |
| *Shortest route distance to school (m)—reference: <500.0* | | |
| 500.0–999.9 | **0.616 [0.385–0.986]** | .043* |
| 1,000.0–1,499.9 | **0.301 [0.188–0.482]** | < .001*** |
| ≥ 1,500.0 | **0.115 [0.073–0.182]** | < .001*** |
| *Average Traffic Risk along Route* | | |
| *Average accident rate per 10⁴ inhabitants crossed—reference: <20.0* | | |
| 20.0–39.9 | **0.636 [0.501–0.807]** | < .001*** |
| ≥ 40.0 | **0.586 [0.397–0.863]** | .007** |
| *Maximum Traffic Risk along Route* | | |
| *Highest accident rate per 10⁴ inhabitants crossed—reference: <20.0* | | |
| 20.0–39.9 | **0.481 [0.317–0.729]** | .001** |
| ≥ 40.0 | **0.315 [0.216–0.459]** | < .001*** |
| *Average Personal Risk along Route* | | |
| *Average crime rate crossed—reference: <90.0* | | |
| 90.0–109.9 | 0.832 [0.553–1.250] | .375 |
| ≥ 110.0 | 1.247 [0.824–1.887] | .296 |
| *Maximum Personal Risk along Route* | | |
| *Highest crime rate crossed—reference: <90.0* | | |
| 90.0–109.9 | **0.592 [0.387–0.906]** | .016* |
| ≥ 110.0 | **0.446 [0.330–0.602]** | < .001** |
| *Average Air Pollution along Route* | | |
| *Average Combined Emission Index crossed—reference: <90.0* | | |
| 90.0–109.9 | 1.218 [0.848–1.750] | .286 |
| ≥ 110.0 | 1.393 [0.816–2.377] | .224 |
| *Maximum Air Pollution along Route* | | |
| *Highest Combined Emission Index crossed—reference: <90.0* | | |
| 90.0–109.9 | **0.600 [0.402–0.894]** | .012* |
| ≥ 110.0 | **0.522 [0.315–0.867]** | .012* |
| *Average Walkability along Route* | | |
| *Average quintile crossed—reference: 1 (least walkable)* | | |
| 2 | 1.019 [0.677–1.534] | .928 |
| 3 | 1.037 [0.636–1.691] | .884 |
| 4 | 1.429 [0.803–2.543] | .225 |
| 5 –Most walkable | **2.611 [1.209–5.639]** | .015* |
| *Minimum Walkability along Route* | | |
| *Lowest quintile crossed—reference: 1 (least walkable)* | | |
| 2 | **1.682 [1.239–2.285]** | .001** |
| 3 | **2.082 [1.371–3.162]** | .001** |
| 4 | **3.994 [2.070–7.703]** | < .001*** |
| 5 –Most walkable | 1.296 [0.266–6.318] | .749 |
| *Average Food Environment Density along Route* | | |
| *Average convenience stores/mile² crossed—reference: ≤20* | | |
| 21–50 | 1.307 [0.663–2.578] | .440 |
| 51–80 | **4.755 [1.973–11.459]** | .001** |

*(Continued)*

**Table 2.** (Continued)

| Variable | Odds Ratio [Confidence interval] | p-value |
|---|---|---|
| **> 80** | **2.813 [1.161–6.820]** | **.022**[*] |
| *Maximum Food Environment Density along Route* | | |
| *Highest convenience stores/mile² crossed—reference: ≤20* | | |
| *21–50* | 0.542 [0.183–1.601] | .268 |
| *51–80* | 0.488 [0.160–1.493] | .209 |
| *> 80* | 0.581 [0.176–1.923] | .374 |

closer to the centre of London, were thus disproportionally exposed to unhealthy nutritional options. This association might be the consequence of an activity-inciting effect of convenience stores, acting as potential intermediate destinations during the school commute, or be caused by the strategic location of such stores in areas with high volumes of active commuters [49]. If the first interpretation is correct, then the potential reduction in actively commuting children could be compensated by offering other incentives to walk or cycle, for instance by reducing air pollution and creating safer road environments. The latter interpretation supports initiatives aimed at reducing the number of convenience stores around schools, and stimulating the healthy food options they offer [50].

## Effect of demographic and socioeconomic characteristics

This study also adds to the knowledge of demographic and socioeconomic factors associated with commuting to school. No sex-difference was found. Prior research on this association has often produced conflicting results [51]. Our findings agree with previous Dutch and US studies [52], but contrast with earlier UK research suggesting boys were more likely to actively commute [53]. The latter study, however, focused specifically on independent mobility, where sex-differences might be more pronounced. Older SLIC children were significantly more likely to actively commute, in keeping with the growing evidence on this relationship across Europe, including the UK [39, 52, 54, 55].

Within our sample, black children had about half the odds of walking or cycling to school in comparison to those in the white/other group. No such difference was found for South Asian children. Earlier UK research pointed out that black African-Caribbean primary school-children were more likely to travel by public transport, and tended to live further away from school than children from other ethnicities [56, 57]. Our findings show this group, predominantly residing in south, east and northeast London, was less likely to walk or cycle independent of proximity or other built environmental and socioeconomic characteristics. These areas should therefore be a prime focus of physical activity interventions. Ethnicity can play both a confounding and mediating role in this relation, as the social environment generated by ethnic residential segregation might shape physical activity behavioural choices [58].

Children in the obese category were less likely to actively commute than those with healthy fat mass levels, although the direction of this relation cannot be conclusively determined. Whilst being the most appropriate measure of adiposity, FMI is rarely used in studies relating body composition to activity. The large majority of earlier studies, predominantly using Body Mass Index and fat mass percentage, obtained conflicting evidence on this relation [8, 59].

Turning attention to socioeconomic status revealed that children from highly affluent families and families owning a car had about half the odds of commuting actively to school compared to those in the least affluent group or without access to a car, a trend widely supported throughout literature [60, 61]. This group of families mainly resided towards the northern

**Table 3. Associations between built environmental characteristics, potential confounders and SLIC children's active commuting.**

| Variable | Wald Chi-Squared = 254.7; p < .001 | |
| --- | --- | --- |
| | Odds Ratio [95% Confidence interval] | p-value |
| **Proximity** | | |
| **Shortest route distance to school (m)—reference: <500.0** | | |
| 500.0–999.9 | **0.615 [0.380–0.995]** | .047* |
| 1,000.0–1,499.0 | **0.295 [0.182–0.479]** | < .001*** |
| ≥ 1,500.0 | **0.117 [0.071–0.192]** | < .001*** |
| **Maximum Traffic Risk along Route** | | |
| **Highest accident rate per $10^4$ inhabitants crossed—reference: <20.0** | | |
| 20.0–39.9 | **0.592 [0.370–0.946]** | .029* |
| ≥ 40.0 | 0.710 [0.418–1.207] | .206 |
| **Maximum Personal Risk along Route** | | |
| **Highest crime rate crossed—reference: <90.0** | | |
| 90.0–109.9 | 1.029 [0.603–1.755] | .916 |
| ≥ 110.0 | 0.861 [0.559–1.327] | .498 |
| **Maximum Air Pollution along Route** | | |
| **Highest Combined Emission Index crossed—reference: <90.0** | | |
| 90.0–109.9 | 1.121 [0.701–1.793] | .634 |
| ≥ 110.0 | 1.121 [0.612–2.053] | .712 |
| **Minimum Walkability along Route** | | |
| **Lowest quintile crossed—reference: 1 (least walkable)** | | |
| 2 | 1.017 [0.713–1.451] | .926 |
| 3 | 0.999 [0.637–1.567] | .996 |
| 4 | 0.964 [0.474–1.963] | .920 |
| 5 –Most walkable | 1.004 [0.186–5.422] | .996 |
| **Average Food Environment Density along Route** | | |
| **Average convenience stores/mile$^2$ crossed—reference: ≤20** | | |
| 21–50 | 1.156 [0.557–2.398] | .697 |
| 51–80 | **3.380 [1.313–8.698]** | .012* |
| > 80 | 1.628 [0.608–4.360] | .332 |
| **Sex (reference: female)** | | |
| Male | 0.973 [0.793–1.193] | .789 |
| **Age at test; years from 5 to 11** | **1.089 [1.022–1.162]** | .009** |
| **Ethnicity (reference: white/other)** | | |
| Black | **0.539 [0.394–0.737]** | < .001*** |
| South Asian | 0.817 [0.601–1.111] | .197 |
| **FMI Weight Status (reference: Normal fat mass)** | | |
| Underweight (<5th percentile) | 1.114 [0.567–2.187] | .755 |
| Overweight (85th-95th percentile) | 0.846 [0.600–1.194] | .342 |
| Obese (≥ 95th percentile) | **0.569 [0.371–0.873]** | .010* |
| **Family Affluence Scale (reference: low)** | | |
| Moderate | 0.816 [0.539–1.235] | .336 |
| High | **0.578 [0.356–0.936]** | .026* |
| **Free school lunches (reference: no)** | | |

(*Continued*)

**Table 3.** (Continued)

| Variable | Wald Chi-Squared = 254.7; p < .001 | |
|---|---|---|
| | Odds Ratio [95% Confidence interval] | p-value |
| Yes | 0.927 [0.685–1.255] | .625 |
| Cars owned (reference: 0) | | |
| 1 | **0.593 [0.443–0.795]** | < .001*** |
| 2 | **0.464 [0.321–0.671]** | < .001*** |
| IMD (reference: low) | | |
| Intermediate | 0.912 [0.651–1.275] | .589 |
| High | 0.757 [0.511–1.123] | .167 |
| Level 2: Variance on School Level | 0.276 [0.094–0.812] | |

fringes of London. Neighbourhood deprivation was not significantly associated with these children's commuting choices. Conflicting associations on the consequences of diverging deprivation levels on children's commuting emerged in prior research. Children residing and attending schools in neighbourhoods of lower socioeconomic status were found to have higher [62], mixed [63] or lower [39, 64] levels of walking and cycling to school and physical activity, depending on the research location and context. Whilst area variation is partly captured by the second level of the multilevel models, our null-findings highlight that family socioeconomic status is likely to be more decisive in shaping commuting choices. Moreover, the UK social housing policy is explicitly aimed at creating socially mixed neighbourhoods [65]. This may reduce socioeconomic residential segregation and thereby reduce the impact of small-scale neighbourhood deprivation. Finally, also here, perceptions of deprivation might be dominant.

## Research implications and limitations

This research has several wider implications. Firstly, it demonstrates that the objective built environment is significantly related to commuting mode choices for this multi-ethnic sample of UK schoolchildren. The predominance of associations with proximity shows that the equal provision of high-quality education across cities might be key in inciting active school transport. Secondly, active commuters are often exposed to hazardous built environments. The associations with traffic and personal safety, air pollution and walkability demonstrate the urgent need to provide children with safe and clean commuting routes. Next, the need for healthy food environments around schools was highlighted. This might incite actively commuting children and their parents to choose healthy food if they shop during the school commute. Finally, the confounder relations demonstrated a need to target health-promoting interventions at specific groups: black children, children from high socioeconomic status backgrounds and those living in families with one or multiple cars.

While the large, multi-ethnic sample, the multilevel modelling approach and the wide variety of included built environmental variables along the route are clear strengths of this study, several limitations also need to be acknowledged. First, the cross-sectional analyses do not allow causality to be established. Next, FEAT data were only available for 2014, the year following the conclusion of SLIC data collection. While no dramatic changes are expected during this year, small inaccuracies in convenience stores densities cannot be ruled out. Third, the self-reported commuting data could not be objectively verified. Fourth, whilst a wide variety of thoroughly tested potential confounders was selected, the inclusion of all relevant individual,

family or neighbourhood variables cannot be guaranteed. Moreover, the assumption was made that children follow the shortest route to school. It was impossible to account for detours or the avoidance of specific road segments which might weaken the relationships with environmental hazards. Next, information on the presence of siblings and peers, parental and school attitudes to physical activity, and distances to public transport, which could influence children's mode of commuting, was unavailable and could therefore not be accounted for. Finally, while the SLIC sample was representative of a UK inner-city population, our results only apply to this set of London schools and pupils.

## Conclusion

Our research shows how objectively measurable characteristics of the London built environment can be used to predict commuting behaviour for a representative sample of primary schoolchildren participating in the SLIC study. Proximity to school is the key characteristic associated with active commuting to school. However, personal and road safety and the provision of a healthy environment in terms of food options and air pollution along the route to school also require the attention of urban planners and policymakers. Specific attention should be given to children from minority backgrounds and affluent families. Further work could now be conducted to study the impact of these factors on conscious commuting decisions.

## Supporting information

**S1 Table. Manuscript dataset Bosch et al.**
(XLSX)

**S2 Table. Descriptive statistics for the sample of SLIC children included in the current study.**
(DOCX)

## Acknowledgments

The authors would like to thank the children participating in the SLIC study and their parents/ guardians, as well as the SLIC team at UCL Great Ormond Street Institute of Child Health, London. This work was supported by the NIHR GOSH BRC. The authors are grateful to the UCL Health and Social Surveys Research Group for their insights into London Walkability Index data (© UCL Street Mobility Project—Fig 5). The authors also would like to thank the developers of the FEAT tool at the University of Cambridge's MRC Epidemiology Unit for their help in understanding the tool (Copyright and database right © 2017 CEDAR/MRC Epidemiology Unit. All rights reserved–Fig 6). The cartography and statistical analyses also contain Ordnance Survey, Open Street Map and Open Government data, subject to copyright (Map data © OpenStreetMap contributors; © Crown Copyright and Database Right 2017. OS (100059028); Contains National Statistics data © Crown copyright and database right 2017; Includes data licensed from PointX © Database Right/Copyright 2017 and OS © Crown Copyright 2017. All rights reserved. Licence number 100034829 –Fig 6 | Contains public sector information licensed under the Open Government Licence v3.0 –Figs 1, 2, 3, 4, 7 and 8).

## Author Contributions

**Conceptualization:** Lander S. M. M. Bosch, Jonathan C. K. Wells, Alice M. Reid.

**Data curation:** Jonathan C. K. Wells.

**Formal analysis:** Lander S. M. M. Bosch.

**Funding acquisition:** Sooky Lum.

**Investigation:** Sooky Lum.

**Methodology:** Lander S. M. M. Bosch, Jonathan C. K. Wells, Alice M. Reid.

**Project administration:** Sooky Lum.

**Resources:** Sooky Lum.

**Software:** Lander S. M. M. Bosch.

**Supervision:** Alice M. Reid.

**Visualization:** Lander S. M. M. Bosch.

**Writing – original draft:** Lander S. M. M. Bosch.

**Writing – review & editing:** Jonathan C. K. Wells, Sooky Lum, Alice M. Reid.

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
