## [Decision Letter · Decision Letter 0]

7 Jan 2020

PONE-D-19-29042

Associations of the objective built environment along the route to school with children’s modes of commuting: a multilevel modelling analysis (the SLIC study)

PLOS ONE

Dear Mr Bosch,

Thank you for submitting your manuscript to PLOS ONE. After careful consideration, we feel that it has merit but does not fully meet PLOS ONE’s publication criteria as it currently stands. Therefore, we invite you to submit a revised version of the manuscript that addresses the points raised during the review process.

We would appreciate receiving your revised manuscript by Feb 21 2020 11:59PM. To enhance the reproducibility of your results, we recommend that if applicable you deposit your laboratory protocols in protocols.io, where a protocol can be assigned its own identifier (DOI) such that it can be cited independently in the future. For instructions see: http://journals.plos.org/plosone/s/submission-guidelines#loc-laboratory-protocols

We look forward to receiving your revised manuscript.

Kind regards,

Francisco Javier Huertas-Delgado, Ph.D.

Academic Editor

PLOS ONE

Journal Requirements:

Reviewers' comments:

Reviewer's Responses to Questions

**Comments to the Author**

1. Is the manuscript technically sound, and do the data support the conclusions?

Reviewer #1: Yes

Reviewer #2: Yes

2. Has the statistical analysis been performed appropriately and rigorously? 

Reviewer #1: Yes

Reviewer #2: Yes

3. Have the authors made all data underlying the findings in their manuscript fully available?

Reviewer #1: Yes

Reviewer #2: Yes

4. Is the manuscript presented in an intelligible fashion and written in standard English?

Reviewer #1: Yes

Reviewer #2: Yes

5. Review Comments to the Author

Reviewer #1: GENERAL COMMENTS

The manuscript is well written and structured in a scientific manner. It increases the knowledge of children’s modes of commuting.

I would suggest the authors only to take into account instead of the large sample of the SLIC project (n=2171), the final sample of this study (n=1.889).

SPECIFIC COMMENTS

L 116-117- The final sample from lines 257-258 should be described instead of the whole SLIC sample. The sex and the age distribution should be mentioned.

Reviewer #2: Dear authors,

Thanks for an interesting and very clearly written manuscript.

Ther were only a few thoughts I would like to raise

1) Where there any differences in the characteristics of the children excluded and in those included in the analysis? Please add a comment / description.

2) Should also the parents' PA / attitude be taken into account as possible confounders in the analyses? What about school scores? I think these could make a difference, too. Was such information available / asked? If not, perhaps they should be mentioned in the discussion as missing pieces of information.

3) Should the number of siblings or possibility to commute with friends / alone or proximity to public transport (tube station / bus stop) be taken into account in the analyses as possible confounders, i.e. should they be mentioned in the dicussion as missing? Did you have such information available / was it asked?

One minor detail in line 326 in page 15. I was wondering whether "or" would be better than "and" (...overweight OR obesity...)

Good luck with the publication.

6. PLOS authors have the option to publish the peer review history of their article (what does this mean?). If published, this will include your full peer review and any attached files.

Reviewer #1: No

Reviewer #2: No

---

## [Author Response · Author response to Decision Letter 0]

16 Jan 2020

Reviewer 1

General Comments

1) The manuscript is well written and structured in a scientific manner. It increases the knowledge of children’s modes of commuting. I would suggest the authors only to take into account instead of the large sample of the SLIC project (n=2171), the final sample of this study (n=1.889).

We thank the reviewer for this observation, and took this forward in our manuscript. To facilitate the interpretation of our data and analyses, we have decided to only focus on the final sample of 1,889 children included in this study, instead of also referring to the entire SLIC cohort of 2,171 children. Reference to the latter has been removed from the ‘Abstract’, ‘Materials and Methods’ and ‘Results’ sections, more specifically on lines 35, 119, 259 and 264 in the revised manuscript.

Specific Comments

2) Lines 116-117: The final sample from lines 257-258 should be described instead of the whole SLIC sample. The sex and the age distribution should be mentioned.

In accordance with the reviewer’s general comment, we described only the final sample, both on lines 116-117 (line 119 in the reviewed manuscript), and lines 257-258 (line 264 in the revised manuscript), as well as on line 35 in the abstract. Moreover, we moved the sex and age distribution information from the ‘Results’ section on lines 334-340 to the ‘Materials and Methods’ section on lines 121-124 in the revised manuscript, as suggested by the reviewer.

Reviewer 2

1) Where there any differences in the characteristics of the children excluded and in those included in the analysis? Please add a comment/description.

Due to the focus on the final sample, as suggested by Reviewer 1, references to the overall SLIC sample and excluded children are no longer included in the revised manuscript. For information, however, the children in the sample withheld for multilevel modelling did not differ significantly in terms of age, sex or ethnicity from the full SLIC sample. The excluded children, however, had a 7% higher proportion of males, were slightly older (by 0.2 years), and had a higher proportion of black students (49% versus 26% in the included sample). The latter is due to the fact that the lion’s share of missing data and consequentially excluded children stemmed from three schools with circa 75% black students.

2) Should also the parents' PA/attitude be taken into account as possible confounders in the analyses? What about school scores? I think these could make a difference, too. Was such information available/asked? If not, perhaps they should be mentioned in the discussion as missing pieces of information.

We agree with the reviewer that this information could have further added to insights provided by our study. However, as these data were unfortunately not collected as part of the SLIC Study, we included this in the limitations in the ‘Discussion’ section on lines 557-560 in the revised manuscript.

3) Should the number of siblings or possibility to commute with friends/alone or proximity to public transport (tube station/bus stop) be taken into account in the analyses as possible confounders, i.e. should they be mentioned in the discussion as missing? Did you have such information available/was it asked?

Similar to the information on parents’ and schools’ attitudes, this information was sadly unavailable for the SLIC sample. Hence, we also included this in the limitations in the ‘Discussion’ section on lines 557-560 in the revised manuscript.

4) One minor detail in line 326 in page 15. I was wondering whether "or" would be better than "and" (...overweight OR obesity...)

 We agree with the reviewer that ‘or’ is better here, and have adjusted this accordingly on line 341 of the revised manuscript.

---

## [Decision Letter · Decision Letter 1]

25 Mar 2020

Associations of the objective built environment along the route to school with children’s modes of commuting: a multilevel modelling analysis (the SLIC study)

PONE-D-19-29042R1

Dear Dr. Bosch,

We are pleased to inform you that your manuscript has been judged scientifically suitable for publication and will be formally accepted for publication once it complies with all outstanding technical requirements.

With kind regards,

Francisco Javier Huertas-Delgado, Ph.D.

Academic Editor

PLOS ONE

---

## [Editor Report · Acceptance letter]

27 Mar 2020

PONE-D-19-29042R1 

Associations of the objective built environment along the route to school with children’s modes of commuting: a multilevel modelling analysis (the SLIC study) 

Dear Dr. Bosch:

I am pleased to inform you that your manuscript has been deemed suitable for publication in PLOS ONE. Congratulations! Your manuscript is now with our production department. 

With kind regards,

on behalf of

Dr. Francisco Javier Huertas-Delgado 

Academic Editor

PLOS ONE